# Influence of Re on the Plastic Hardening Mechanism of Alloyed Copper

**DOI:** 10.3390/ma16165519

**Published:** 2023-08-08

**Authors:** Mariusz Krupiński, Beata Krupińska, Robert Chulist

**Affiliations:** 1Department of Engineering Materials and Biomaterials, Faculty of Mechanical Engineering, Silesian University of Technology, 18a Konarskiego St., 44-100 Gliwice, Poland; mariusz.krupinski@polsl.pl; 2Institute of Metallurgy and Materials Science of Polish Academy of Sciences, 25 Reymonta St., 30-059 Krakow, Poland; r.chulist@imim.pl

**Keywords:** copper alloys, rhenium, heat treatment, cold plastic treatment

## Abstract

In this paper, we investigated the effect of adding rhenium to Cu-Ni-Si alloys on the mechanical properties and electrical conductivity of these alloys. The scientific objective was to analyze the effect of Re addition on the microstructure of heat- and cold-treated CuNi2Si1 alloys. Transmission electron microscopy (TEM, STEM) and scanning electron microscopy (EDS, WDS) were used to examine the microstructure. Orientation mapping was also performed using a scanning electron microscope (SEM) equipped with a backscattered electron diffraction (EBSD) system. In addition, hardness at low load and conductivity were tested. The obtained results showed that modifying the chemical composition of Re (0.6 wt%) inhibits the recrystallization process in the CuNi2Si1 alloy, which was cold deformed and then subjected to recrystallization annealing.

## 1. Introduction

Intense industrial development has forced the use of alloys with increasingly better and more stable structures and properties, especially at elevated temperatures. To this end, the chemical composition of metal alloys has been modified. Known and commonly used functional materials from the group of release hardened copper alloys, including Cu-Fe, Cu-Cr, Cu-Co, Cu-Ni-Si, and Cu-Ni-Si-Cr, are characterized by good strength properties and insufficient conductivity. The intense development of technologies, for which durability and operational reliability have become a priority, has enforced the need to search for materials with more favorable and more specific functional properties and with high and, above all, stable mechanical, electrical, and thermal characteristics [1,2,3,4,5,6,7,8,9,10].

Using Re as a modifier in metal alloys in traditional metallurgical processes offers advantages, which are important because it does not involve any additional technological costs. Rhenium is harvested during the mining of copper. And, as research shows, even a small amount of the Cu alloy causes significant changes without the need to significantly increase the temperature in the liquid state. This is important for economic reasons and to prevent overheating of the alloy [11,12,13,14].

To obtain excellent properties at very high temperatures and good corrosion properties, alloys of various metals are often used, including iron, nickel, or cobalt in combination with a wide range of rare earth metals. They are mainly applied in aviation, rocket, and space industries, where the elements made from them work under extreme conditions [15,16,17,18].

The dynamic advancement of modern technologies forces the development of materials engineering, necessitating the continuous improvement of existing materials and the development of new materials. The requirements for the materials used are becoming increasingly demanding and more precise. These demands concern the whole spectrum of properties at the same time. Modern materials are expected to be durable, reliable during operation, and possess good strength, electrical, and thermal properties [1,2,3,4,19].

With the development of both new technologies and new research opportunities, the creation of materials is becoming easier. Copper alloys are examples of such materials [20,21,22,23,24,25].

A significant number of the properties of copper alloys, such as Cu-Fe, Cu-Cr, Cu-Co, Cu-Ni-Si and Cu-Ni-Si-Cr, especially their high strength, high conductivity, and thermal stability, are achieved through precipitation hardening [1,2,3,5,6,7,8,9,10]. The CuNi2Si1 alloy, whose mechanical properties can be shaped by heat and plastic treatment, is increasingly becoming a subject of research [5,6,7,8,9,10,18]. It is common to use Cr or Cr, Mg or Cr, and Ti as modifiers in the Cu-Ni-Si alloys. Very often during heat or plastic treatment, the strength properties increase, but the electrical properties deteriorate [1,2,3,4,11]. However, some reports show a change in morphology, which relates to better mechanical properties along with favorable changes in conductivity [1,2,3,4,11,15,20].

According to [1,11,21], Cu-Ni-Si alloys that supersaturate at 950 °C show a wide range of changes in their mechanical properties and, in some parts, also exhibit changes in their electrical conductivity. These changes are closely related to the microstructure formed as a result of heat and plastic treatment. Zhao et al. [7] demonstrated the significant role of nanometric coherent precipitates formed as a result of the supersaturation and aging of CuNiSi alloys. These precipitates (Ni_2_Si) increase with aging time and the distance between them also increases. Changes in the morphology of precipitates in the matrix are associated with changes in their mechanical properties and, in part, changes in electrical conductivity [7]. The precipitation of Ni_2_Si has an intense nature during aging and proceeds homogeneously in the matrix within a temperature range of 267 to 381 °C [7].

Lu et al. [17] showed that there are three two-phase and two three-phase areas in the Cu-rich areas of the isothermal phase system at 300–600 °C. The three two-phase areas are FCC-Al (Cu-rich) + γ-Ni_5_Si_2_, FCC-Al (Cu-rich) + γ-Ni_2_Si, and FCC-Al (Cu-rich) + ε-Ni_3_Si_2_. The two three-phase areas are FCC-Al (Cu-rich) + γ-Ni_5_Si_2_ + δ-Ni_2_Si and FCC-Al (Cu-rich) + δ-Ni2Si + ε-Ni_3_Si_2_. For this reason, the alloy in the Cu-rich part can precipitate ε-Ni_5_Si_2_, δ-Ni_2_Si, or ε-Ni_3_Si_2_. The proportion of each phase depends on the composition of the alloy and the aging temperature. The transmission electron microscope (TEM) analysis of the CuM_12_Si_0.75_ alloy indicates mainly ε-Ni_2_Si and only a few phase particles of γ-Ni_5_Si_2_, which agrees with thermodynamic phase equilibrium calculations.

Rhenium is one of the rare earth metals added to superalloys. The addition of a few percent to the alloy improves virtually all the properties that are important in aerospace applications. The basic effect of Re is to inhibit recrystallization of the alloy grains, which can lead to an increase in the average grain size. This, in turn, resulted in an increase in the brittleness of the alloy during aging at high temperatures. This effect is most likely related to the high ability of this metal to form clusters, which, in the “nano” size, inhibits the migration of other atoms without destroying the homogeneity of the alloy at the micro level. This phenomenon is termed the “rhenium effect” by engineers.

In the industry, alloys with the addition of Re in the range of about 3–6% are very popular. Superalloys of the second generation contain as much as 5–6%. At present, it has been possible to maintain the properties of superalloys with as little as 1.5%. This reduction in Re content is due to the limited availability of this element [15,16,17,18,26,27,28,29,30].

As a result of adding Re to the copper alloy, the following observations were made: (I) Re segregates during the crystallization of the alloy; (II) during solution homogenization, Re is incorporated into the α matrix as a result of its limited solubility in the solid state; (III) Re enhances the strength properties at elevated temperatures and preserves the appropriate conductivity for this type of alloy; and (IV) the precipitation of the Re phase, resulting from the limited solubility of Re in the Cu matrix, blocks the recrystallization process during annealing.

## 2. Materials and Methods

Cu-Ni-Si alloys with the chemical compositions listed in Table 1 were prepared for the study.

After modification with rhenium (0.6 wt%), the CuNi2Si1Re0.6 alloy was heat-treated at 950 °C. Then, the alloy was cold-plastic-deformed with a 50% reduction ratio. After cold working, recrystallization annealing was performed at 450, 500, 550, and 600 °C. The parameters of heat and mechanical treatment are listed in Table 2.

Both heat treatment and cold plastic strain were performed using a DSI (Dynamic System Inc., Austin, TX, USA) Gleeble 3800 thermo-mechanical simulator. The Gleeble 3800 simulator is equipped with a direct resistance heating system that accurately maintains the desired temperature with an accuracy of ±1 °C. The use of the Gleeble simulator allowed heat treatment to be performed at a well-defined temperature, thereby minimizing errors during the heating, annealing, and direct cooling of pressurized water or compressed air jets during the process. To protect the samples from unintentional contact with the tungsten carbide anvil and to improve the contact between the contacting surfaces (which is very important in the case of resistive heating conducted in the simulator, involving the flow of current through the test sample), as well as friction between the connecting surface of the test sample and the surface of the tungsten carbide anvil, a set of graphite and tantalum layers with a thickness of 0.1 mm was used. Additionally, the contact surfaces of the sample and the anvils were coated with nickel-based grease.

In the first stage of heat treatment, consisting of supersaturation of the investigated alloys, cylindrical samples with dimensions of Ø 12 × 12 mm (Figure 1) were resistance-heated in an argon atmosphere at a rate of 3 °C·s^−1^ to a set temperature of 950 °C for both the CuNi2Si1Re0.6 and CuNi2Si1 alloys. The samples were then annealed at the appropriate temperature for 1 h and cooled with a 50 PSI (350 kPa) water jet. Water was sprayed from four nozzles directly onto the sample for 20 s, which allowed the sample to cool to room temperature (approximately 20–30 °C). After supersaturation, the samples were subjected to an intense process of cold plastic strain at a very high strain rate of 100 s^−1^. The strain value was 50% (the sample height before the strain was 12 mm, the sample height after the strain was 6 mm, and the strain = 0.69 (denoted as {ln (end length/initial length)})). The cold plastic strain, which focused on inducing large deformation in the structure of the investigated alloys, was carried out at room temperature (around 23 °C) in a protective argon atmosphere. In the next stage, the samples were annealed from the initial temperature of 450 °C, and the rate of heating the samples to the set temperature was also 3 °C·s^−1^. The annealing time was 1 h. The annealing process was carried out in a protective argon atmosphere. Before each heat treatment or cold plastic strain experiment, a negative pressure of 0.1 mBar was created in the chamber. Subsequently, protective gas was introduced into the working chamber of the Gleeble 3800 simulator. This was followed by the introduction of another protective gas (argon) with a vapor pressure of about 200 mBar.

The microstructure and chemical composition were tested through EDS microanalysis using the Zeiss Supra 25 (Thornwood, NY, USA) and MEF4A scanning electron microscope with attachments. An alloy structure analysis was performed using both a ZEISS optical microscope and image analysis software. The samples prepared for microstructure observation were first mechanically ground and polished. Then, they were electropolished and etched using either an electrochemical reagent or a reagent consisting of ferric chloride, hydrochloric acid, and ethyl alcohol. The structural studies and phase identification of the thin films were performed using transmission electron microscopy (FEI TITAN TEM) at a 300 kV accelerating voltage with electron diffraction (selected area diffraction, SAD). To identify the crystalline phase structures, the microstructural, chemical, and phase compositions were also analyzed using FEI TITAN TEM with SAD (FEI Company, Hillsboro, OR, USA) at a 300 kV accelerating voltage. The resulting diffraction images were analyzed using specialized software designed to resolve electron diffraction images. The preparations for examination in a high-resolution transmission microscope were prepared using a FEI Helios PFIB SEM/Xe-PFIB Microscope. The samples were prepared by applying a protective layer of platinum, after which they were cut using a focused xenon plasma ion beam (Xe-PFIB). Orientation mapping was performed using an FEI Quanta 3D field emission gun scanning electron microscope (SEM) equipped with a TSL electron backscattered diffraction (EBSD) system. The hardness test was conducted using a Vickers Future-Tech hardness tester (FM-ARS 9000, Future-Tech, Tokyo, Japan), with a load of 1000 gf applied for 10 s. Electrical conductivity was measured using a Sigmatest Foerster.

## 3. Results and Discussion

### 3.1. Structure and Phase Composition Analysis

The CuNi2Si1 alloy with a 0.6% rhenium mass solidified into a single α phase at 1083 °C. The phases enriched with nickel, silicon, and rhenium solidified mostly at the α phase grain boundary (Figure 2a). As the end of solidification occurs at 1014 °C [4,5], the supersaturation process was performed at 950 °C for 1 h, resulting in the dissolution of the intermetallic phases into the α phase matrix. After the cold strain with a 50% reduction degree, the microstructure was plastically deformed with numerous slip bands (Figure 2b). In the next step, the sample was subjected to heat treatment. As a result of annealing at 450 °C for 1 h, the rhenium dissolved in the matrix in the form of 360 to 650 nm phases within the α matrix grains (Figure 2c). The literature indicates that recrystallization already occurs at 450 °C for CuNi2Si1 alloys [1,2,3,31,32,33]. However, for rhenium-modified alloys, recrystallization is usually blocked at this temperature.

This study also shows that the Ni_2_Si phases separated within the α matrix (Figure 3a). The EDS chemical composition analysis is shown in Figure 3b. The morphology and size of the rhenium precipitates was first confirmed using scanning electron microscopy. The distribution and morphology of the rhenium phases are shown in Figure 4a, with precipitates in Figure 4b. Transmission electron microscopy studies confirm the occurrence of rhenium phases in the supersaturated, deformed, and subsequently aged alloy. This can be seen in Figure 5a, which shows the dark field images, along with marked points indicating the chemical composition analysis (STEM) in Figure 5b.

The analysis plots at the investigated points are shown in Figure 5c,d. In area 1, the matrix was examined, and the analysis indicates that the alloying elements (Ni, Si, and Re) did not dissolve in the matrix. In area 2, the analysis indicates the presence of the rhenium phase. Diffraction studies were performed to confirm that these were Re phases (Figure 6).

The investigations have revealed the presence of Re-phase particles that occurred in the material after the treatment process. The revealed phases appeared as round, globular particles that were up to 800 nm in diameter. The diffraction pattern calculation gives a clear indication of the Re phase in the metal matrix after alloy treatment (Figure 6a,b). The occurrence of globular Rhenium particles due to the chosen treatment process may be responsible for the strengthening of the material and the increase in the mechanical properties of the obtained in situ composite. The confirmed phase has a hexagonal crystallographic structure within the space group 194—(P 63/mmc) with the [1,2,3,4,5,6,7,8,9,10,11] zone axis (Figure 6c).

The application of 1 h of annealing at 450 °C after cold plastic strain deformation did not make it possible to restore the recrystallized crystal structure of the cold-plastic-deformed CuNi2Si1Re0.6 alloy. The annealing temperature was chosen so that the process could take place above the recrystallization temperature, which was consistent with reports in the literature for this type of CuNi2Si1 alloy [1,3,12,31,32,33]. Recrystallization temperature is a conventional concept and cannot be unambiguously determined as it depends on many factors, such as the melting temperature, alloy composition, the amount of energy stored during the cold plastic strain process, and the annealing time. The recrystallization temperature is most often defined as 0.35–0.6 of the absolute melting temperature, i.e., for the alloys investigated, assuming that the melting temperature is about 1083 °C CuNi2Si1Re0.6, the recrystallization temperature is in the range of 380 °C to 650 °C. Theoretically, the higher the degree of reduction, the lower the recrystallization temperature should be. In the case of the investigated alloys, despite using an annealing temperature of 450 °C (for 1 h), which is close to the upper limit of the recrystallization temperature range, it did not suffice to remove the effects of the reduction and restore the original structure. This may be influenced by alloying additives and admixtures of foreign atoms in the alloy solution, which are released from the solution and inhibit the recrystallization process. In the case of the investigated alloys, the presence of Ni_2_Si (Figure 3a) and the Re phases were found, among others, which exhibited the morphology of the oval phases visible in Figure 4a.

The phase analysis and grain crystallographic orientations (EBSD) using scanning electron microscopy, along with the crystallographic orientation distribution map and texture, are shown in Figure 7 and Figure 8.

The orientation map shows the microstructure with deformed grains after compression with a 50% reduction rate. The microstructure is bimodal with smaller and larger grains; however, both are strongly deformed as they indicate a strong orientation gradient within the individual grains. In addition, the grains have a texture in which the {111} plane is oriented perpendicular to the compression direction. A very similar microstructure was obtained after annealing; however, the grain size was smaller. The obtained grain refinement appears to be linked to the recrystallization process, which was inhibited from the very beginning. In addition, the texture image seems to be more representative of the compression test, as it detected smaller grains.

The observed structure of the cold-strained and aged alloy resulted from the precipitation of Re phases in the structure, which was facilitated by structural defects, such as dislocations, blocking the recrystallization process. These processes strengthened the alloy as a result of plastic strain and the precipitation of the strengthening phases in the alloy. Supersaturation at 950 °C caused the diffusion of Re, and then plastic deformation caused the alloy to harden. The precipitation of the Re phase, resulting from the limited solubility of Re in the Cu matrix, blocked the recrystallization process during annealing.

### 3.2. Conductivity and Hardness Test Results

As a result of the precipitation of the strengthening phases, the hardness of the CuNi2Si1Re0.6 alloy after treatment amounted to 225 HV, which is 50% higher than that of the CuNi2Si1 alloy. After heat and plastic treatment, conductivity in the CuNi2Si1Re0.6 alloy decreased by approximately 15% compared to the alloy without Re addition. The baseline conductivity of the CuNi2Si1Re0.6 alloy was slightly higher than that of the CuNi2Si1 alloy. Changes in hardness at low load force and conductivity are shown in Figure 9 and Figure 10, where it can be observed that the alloy with the addition of Re, after heat treatment and plastic deformation, exhibited greater hardness (by about 75 HV) and lower conductivity (by about 2 Ms/m).

In future studies investigating the alloys, the effect of heat treatment time and temperature on the coherence of the Re phases emitted from the matrix during processing will be addressed. Additionally, the morphology of the microstructure will be described numerically using computer image analysis and statistical methods.

## 4. Conclusions

The performed tests on Cu-Ni-Si alloys with the addition Re at a 0.6% mass level showed that:(I)the addition of rhenium at a 0.6% mass level resulted in a 50% increase in hardness (the hardness of the cold-work-treated CuNi2Si1 alloy was 150 HV, and after the addition of rhenium, it increased to 225 HV);(II)the electrical conductivity of the cold-worked and Re-modified copper alloy was 14 MS/m, which was about 15% lower than the alloy without Re modification;(III)annealing of the CuNi2Si1Re0.6 alloy, previously solution-saturated and after cold working, caused separation of the Re phases of about 350–550 nm in the α phase matrix;(IV)the modification of the chemical composition with rhenium caused the fragmentation of the microstructure and blocked recrystallization mechanisms, which, for the alloy without rhenium addition, occurs already at about 450 °C.

Despite the insolubility of rhenium in copper in the solid state, supersaturation increases the degree of homogenization. During supersaturation, diffusion processes take place. Mass transport causes the formation of nanometric Re phases in the α matrix. At the same time, due to the high activation energy, the resulting Re phases block the recrystallization mechanisms. The Re phases increase the recrystallization temperature (at the assumed degree of deformation) to larger ranges compared to Cu-Ni-Si alloys, where the recrystallization temperature is 450 °C, as confirmed by other test results [1,6].

## Figures and Tables

**Figure 1 materials-16-05519-f001:**
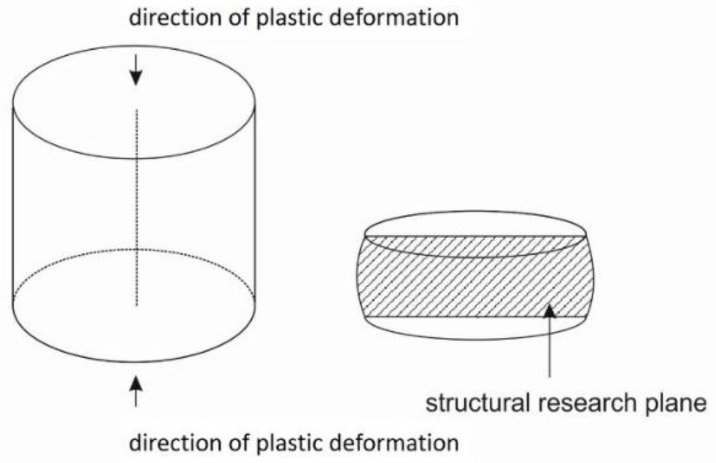
Cylindrical samples with dimensions of Ø 12 × 12 mm.

**Figure 2 materials-16-05519-f002:**
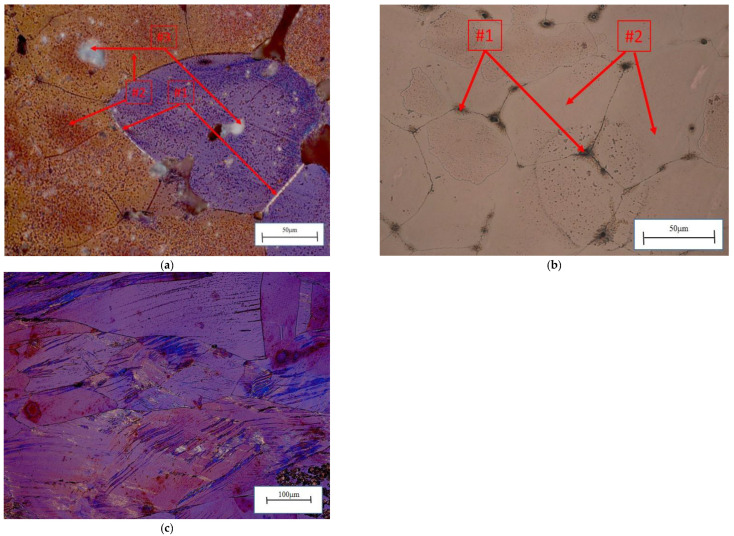
(**a**) Microstructure of the CuN2iSi1Re0.6 alloy, initial state (#1—Ni_2_Si phases; #2—α phases; #3—Re phases); (**b**) structure of the CuNi2Si1 alloy, initial state (#1—Ni_2_Si phases; #2—α phases;); and (**c**) structure of the CuNi2Si1Re0.6 alloy (supersaturation at 950 °C, (Ar); 1 h of cooling in water; cold plastic deformation 50%; annealing at 450 °C for 1 h).

**Figure 3 materials-16-05519-f003:**
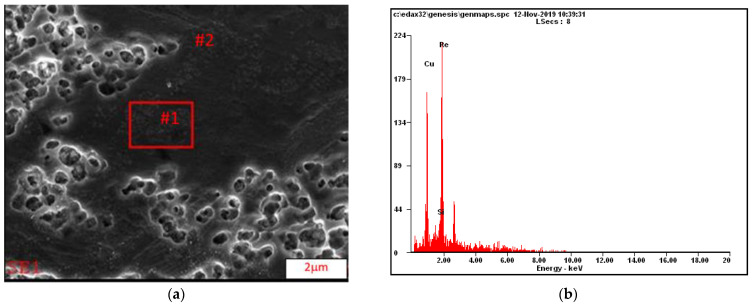
(**a**) Microstructure of the CuNi2Si1Re0.6 alloy (supersaturation at 950 °C, (Ar); 1 h of cooling in water; cold plastic deformation 50%; annealing at 450 °C for 1 h) (#2—Ni_2_Si phases) and (**b**) EDS analysis from micro area #1 (Table 3).

**Figure 4 materials-16-05519-f004:**
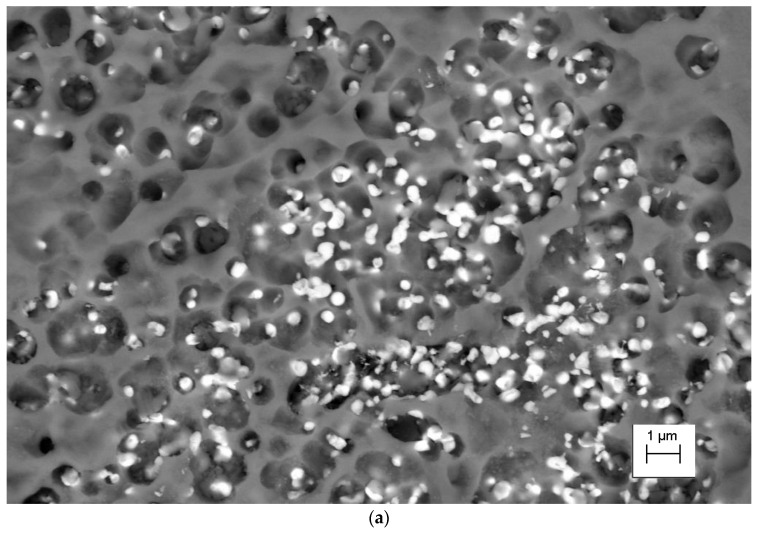
(**a**,**b**) Structure of the CuNi2Si1Re0.6 alloy (supersaturation at 950 °C, (Ar); 1 h of cooling in water; cold plastic deformation 50%; annealing at 450 °C for 1 h).

**Figure 5 materials-16-05519-f005:**
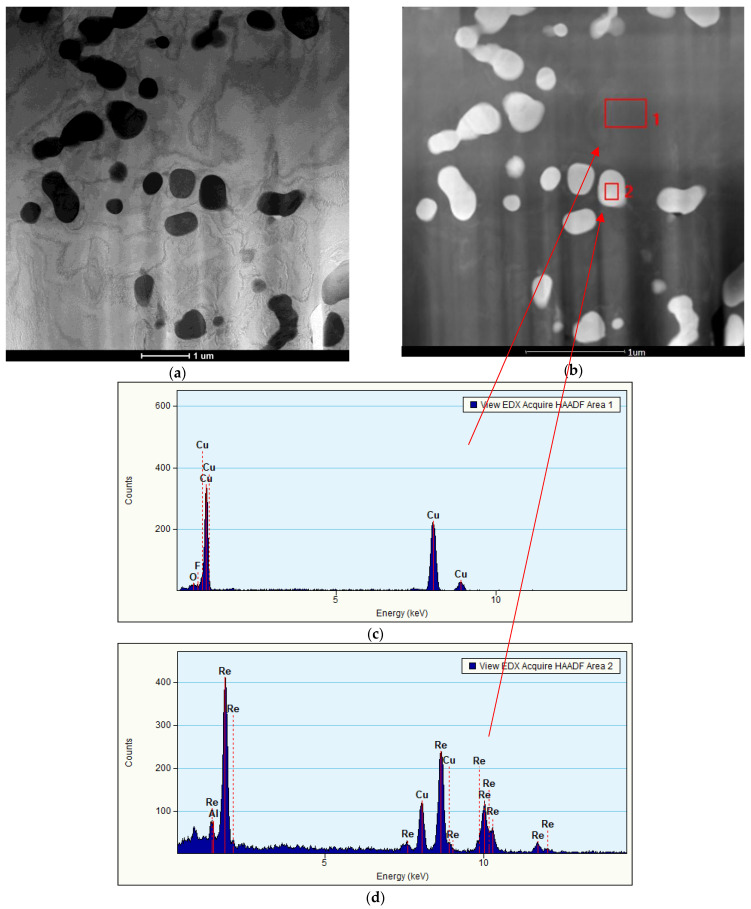
(**a**) Microstructure of the CuNi2Si1Re0.6 alloy (supersaturation at 950 °C, (Ar); 1 h of cooling in water; cold plastic deformation 50%; annealing at 450 °C for 1 h), dark field image; (**b**) structure of the CuNi2Si1Re0.6 alloy (supersaturation at 950 °C, (Ar); 1 h of cooling in water; cold plastic deformation 50%; annealing at 450 °C for 1 h; (**c**) analysis of zone 1 made with energy-dispersive X-ray spectroscopy; and (**d**) analysis of zone 2 made with energy-dispersive X-ray spectroscopy.

**Figure 6 materials-16-05519-f006:**
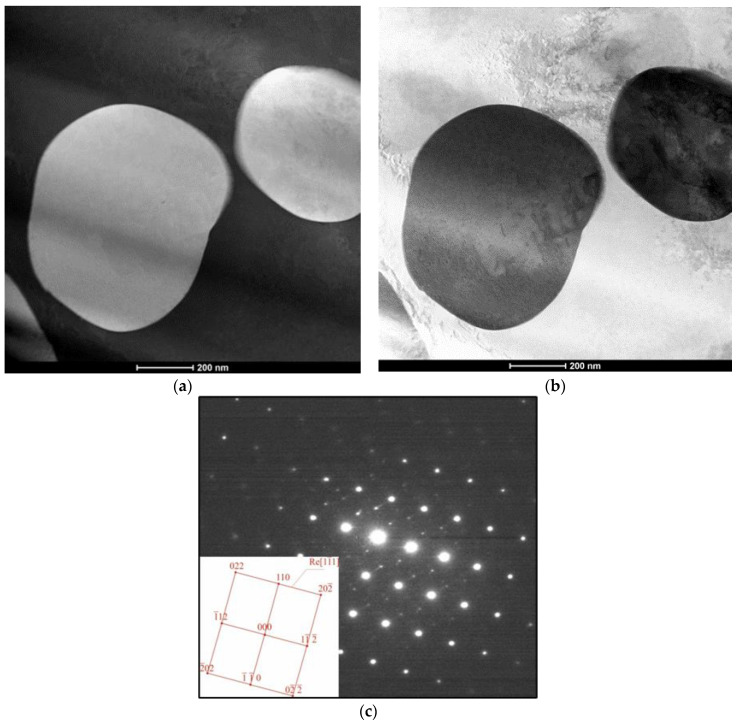
Structure of the CuNi2Si1 alloy modified with Re: (**a**) light field image, (**b**) dark field image, and (**c**) the diffraction pattern of zone axis for [1,2,3,4,5,6,7,8,9,10,11] Re.

**Figure 7 materials-16-05519-f007:**
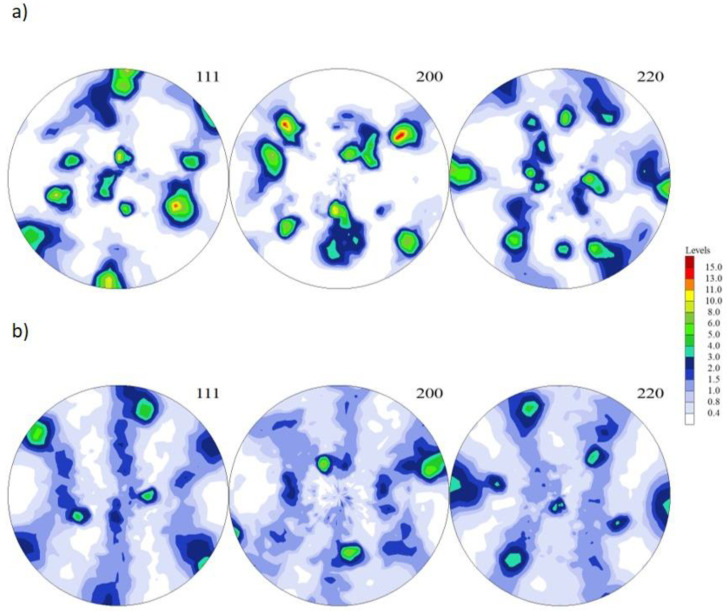
(**a**) (111), (200), and (220) pole figures of the CuNi2Si1Re0.6 alloy (supersaturation at 950 °C, (Ar); 1 h of cooling in water; cold plastic deformation 50%) and (**b**) structure of the CuNi2Si1Re0.6 alloy (supersaturation at 950 °C, (Ar); 1 h of cooling in water; cold plastic deformation 50%; annealing at 450 °C for 1 h).

**Figure 8 materials-16-05519-f008:**
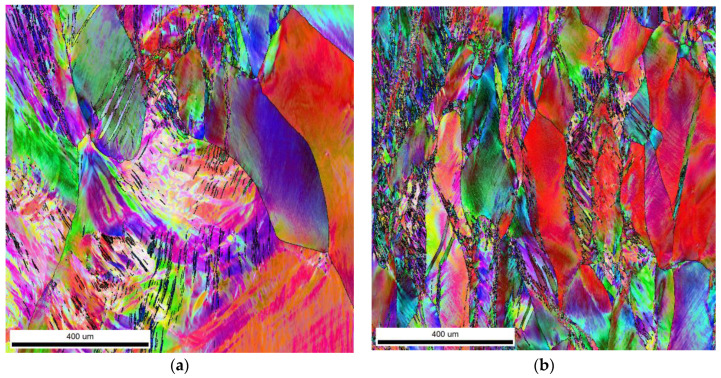
(**a**) Electron backscatter diffraction (EBSD) mapping of CuNi2Si1Re0.6 (supersaturation at 950 °C, (Ar); 1 h of cooling in water, cold plastic deformation 50%) and (**b**) electron backscatter diffraction (EBSD) image structure of the CuNi2Si1Re0.6 alloy (supersaturation at 950 °C, (Ar); 1 h of cooling in water; cold plastic deformation 50%; annealing at 450 °C for 1 h).

**Figure 9 materials-16-05519-f009:**
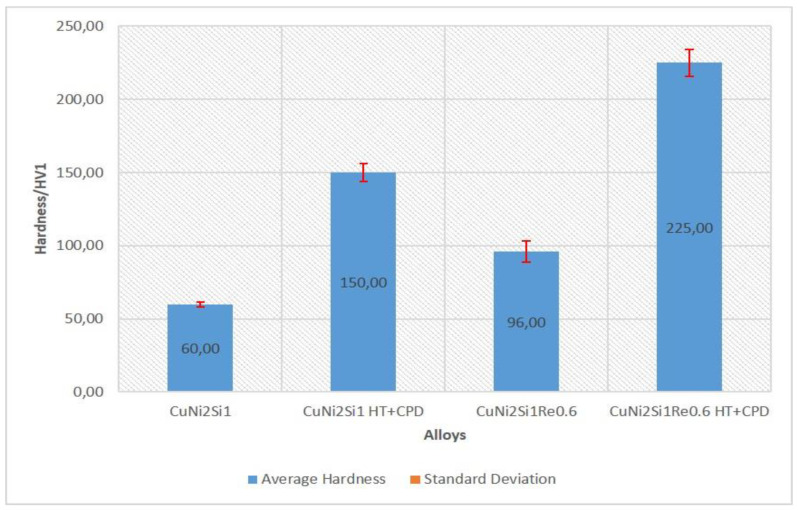
Results of the hardness (HV) test for the structure of the CuNi2Si1Re0.6 alloy in the cast state and the structure of the CuNi2Si1Re0.6 alloy (supersaturation at 950 °C, (Ar); 1 h of cooling in water; cold plastic deformation 50%; annealing at 450 °C for 1 h). HT + CPD—after heat treatment plus cold plastic deformation.

**Figure 10 materials-16-05519-f010:**
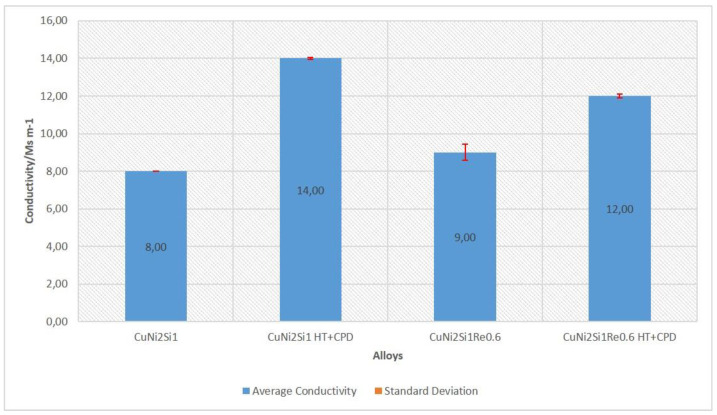
Results of the electrical conductivity (γ) test for the structure of the CuNi2Si1Re0.6 alloy in the cast state and the structure of the CuNi2Si1Re0.6 alloy (supersaturation at 950 °C, (Ar); 1 h of cooling in water; cold plastic deformation 50%; annealing at 450 °C for 1 h). HT + CPD—after heat treatment plus cold plastic deformation.

**Table 1 materials-16-05519-t001:** Chemical compositions of the investigated copper alloys.

Alloys	Elements as Compounds of the Modeled Cu Casts, Mass%			
	Ni	Si	Re	Cu
Cu-Ni-Si	2	1	-	rest
Cu-Ni-Si-Re	2	1	0.6	rest

**Table 2 materials-16-05519-t002:** Heat treatment conditions.

	Supersaturation	Plastic Deformation	Annealing
Heat treatment temperature	950 °C	room temperature	450 °C
time	1 h	-	1 h
cooling rate	20 s	-	-
strain rate	-	100 s^−1^	-

**Table 3 materials-16-05519-t003:** Results of the EDS spectrum analysis for the areas from Figure 3.

Element	Area #1, *% wt.*	Area #1, *% at.*
Ni	1.97	4.34
Si	4.26	4.48
Cu	93.77	91.18

## Data Availability

Not applicable.

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
