# Peer review of "Influence of Re on the Plastic Hardening Mechanism of Alloyed Copper"

_materials, 2023, doi:10.3390/ma16165519_

Round 1

Reviewer 1 Report

In the paper, the effect of rhenium addition to Cu-Ni-Si alloys on mechanical properties and electrical conductivity was investigated. The scientific objective was to analyze the effect of the Re addition on the microstructure of heat-and cold-treated CuNi2Sil alloys. According to the journal's standards, some parts of this work should be revised according to the journal's criteria.

1) In Chapter 3.2, the author writes, "After the heat and plastic treatment, conductivity in CuNi2Si1Re0.3 alloy decreases by approximately 15% than in the alloy without Re addition. ", CuNi2Si1Re0.3 does not appear to be used in the article.

2) In Conclusion (III), how to calculate the precipitation size of Re phases in α phase? How much area was the data collected?

3) The manuscript and conclusion of the article repeatedly compare the effect of adding Re element on the microstructure of the alloy, but there is no microstructure of the alloy without adding Re element. What is the basis of the conclusion (iv)? Please explain.

4) The article only compares the microstructure changes of materials before and after annealing. The research of annealing can enhance the properties of materials has been widely studied. What is the significance of this work? Please indicate.

5) Where the hardness test was performed on the sample and how many times the hardness tested? The standard error in the hardness chart is recommended to be replaced by an error bar.

6) Only the hardness to characterize the mechanical properties of the material is not comprehensive enough, the authors can add tensile or compression test to characterize the strength of the material, after the grain refinement, the strength and toughness of the material should have significant changes.

7) The quality of Figures 1,2 4 and 5 should be improved. In Figure 5, It is recommended to use a unified scale bar.

8) There are some grammar error, which should be improved.

For example, in Chapter 3, "The EDS chemical composition analysis is shown in Fig. 2b.” It should be changed to "The EDS chemical composition analysis is shown in Fig. 3b.”

Minor editing of English language required

Reviewer 2 Report

After reviewing the manuscript, I have the following questions:

1) The introduction is not concise, which is difficult for readers to understand why the authors add the Re element into the Cu-Ni-Si alloy.

2) The authors stated that without Re addtion the recrystallization temperature of the CuNi2Si1 alloy is 450 ℃. In the manuscript, however, there is no evidence of the recrystallized structure for comparison (there is no microstructure of CuNi2Si1 alloy at all while it was mentioned in the experimental section).

3) The plastic hardening mechanism was not clearified by only testing micro-hardness variation.

4) All the figures are blurred, some fonts are even difficult to read.

5) The references are not up to date, considering that there are so many investigations on Cu-Ni-Si alloys recently.

6) Some typos were detected.

Some sentences are difficult to understand and should be improved.

Reviewer 3 Report

Article title: Influence of Re on the plastic hardening mechanism of alloyed-copper (materials-2529354)

  This paper presents a study on the influence of rhenium on copper-based alloys. The subject is interesting and the types of analysis performed are appropriate. However, there are many mistakes or unclear sentences and so a strong revision is mandatory.

Page 2 line 55

This sentence is not clear maybe it should be better to change and integrate it with the other sentences around.

Page 2 line 58

This sentence is too vague. Which reports are you referring to?

Section 3.1

In this section, the figure numbering appears to be incorrect. Some examples:

Line 175 

Fig. 2b or maybe 3b

Line 177

Fig. 3a or maybe 4a

Line 178 

Fig 4b is not an image of precipitate but it is an EDS graph!

Regarding the figures some of the are not readable: fig 4b and figs 5c and 5d. The graphical quality must be improved.

In my opinion, some labels should be added to figure 2a in order to identify the different zones and their changes in figures 2b and 2c.

Round 2

Reviewer 1 Report

Please improve the quality of Figures 9 and 10. They are not clear.

Reviewer 2 Report

The quality of the figures is very low. Please improve all the figures if they are not technical issues due to the Journal.

None.